# Natural Polyphenols Targeting Senescence: A Novel Prevention and Therapy Strategy for Cancer

**DOI:** 10.3390/ijms21020684

**Published:** 2020-01-20

**Authors:** Yan Bian, Juntong Wei, Changsheng Zhao, Guorong Li

**Affiliations:** Shandong Provincial Key Laboratory of Animal Resistant, School of Life Sciences, Shandong Normal University, Jinan 250014, Shandong, China; 2017020748@stu.sdnu.edu.cn (Y.B.); 2017020759@stu.sdnu.edu.cn (J.W.); 2017020758@stu.sdnu.edu.cn (C.Z.)

**Keywords:** polyphenol, cell senescence, cancer inhibition, tumor microenvironment

## Abstract

Cancer is one of the most serious diseases endangering human health. In view of the side effects caused by chemotherapy and radiotherapy, it is necessary to develop low-toxic anti-cancer compounds. Polyphenols are natural compounds with anti-cancer properties and their application is a considerable choice. Pro-senescence therapy is a recently proposed anti-cancer strategy and has been shown to effectively inhibit cancer. It is of great significance to clarify the mechanisms of polyphenols on tumor suppression by inducing senescence. In this review, we delineated the characteristics of senescent cells, and summarized the mechanisms of polyphenols targeting tumor microenvironment and inducing cancer cell senescence for cancer prevention and therapy. Although many studies have shown that polyphenols effectively inhibit cancer by targeting senescence, it warrants further investigation in preclinical and clinical studies.

## 1. Introduction

Cancer, a disease with a high incidence and mortality, has become one of the most significant health problems. From 2016 to 2017, cancer cases increased from 17.2 million to 24.5 million while the cancer death increased from 8.9 million to 9.6 million worldwide. In 2018, there were an estimated 18.1 million new cancer cases (17.0 million, except for non-melanoma skin cancer) and 9.6 million people died of cancer (9.5 million, except for non-melanoma skin cancer) in 185 countries [1,2,3], which indicates that there are still huge challenges in cancer treatment. Because of the universal non-specific effects and inevitable toxicity of cancer drugs nowadays, tissues adjacent to cancer will be implicated and the normal cell will be killed simultaneously when cancer cells are scavenged. It is of great significance to discover or design new agents with low toxicity to normal cells and effective therapy of cancer. Increasing the number of natural compounds has been highly valued for their anti-cancer properties. Seeing that compounds derived from natural sources are easily available, inexpensive, biologically active, and mildly toxic, they are expected to be potential chemotherapeutic drugs in the future. Polyphenolic compounds, generally discovered from plants, are synthesized to adapt the stressful and pathogenic environmental situations [4]. They are characterized by more than one connected hydroxylated/multi-hydroxylated benzene ring-phenol, and are classified into four different groups based on their structural properties: phenolic acids, stilbenes, lignans, and flavonoids [5]. Currently, the anti-tumor profile of polyphenols puts them in the spotlight of research fields. Polyphenols exert their potential in anti-cancer therapy via promoting apoptosis, regulating autophagy, inhibiting proliferation and migration, and so on. The mechanisms include epigenetic modulation as well. For instance, plant polyphenol curcumin regulated miRNA related to tumorigenesis, induced global hypomethylation, and decreased histone deacetylases (HDACs) expression to inhibit the cell proliferation, metastasis, angiogenesis, and induce apoptosis in cancer cells [6,7,8,9]. Oleacein, one of olive oil polyphenols, promoted histone acetylation, inhibited clonogenicity, induced cell cycle blockade, and apoptosis in human multiple myeloma cells [10,11]. Epigallocatechin-3-gallate, a major component of green tea polyphenols, downregulated expression of human telomerase reverse transcriptase (hTERT), a major enzyme determining telomere stability, through causing promoter hypomethylation and histone deacetylations, thereby inhibiting proliferation of breast cancer cells [12]. Polyphenols extracted from Annurca apple selectively inhibited the viability of MDA-MB-231, a triple negative human breast carcinoma cell line, by causing G2/M phase arrest, inducing intrinsic and extrinsic apoptosis and beclin-1-independent autophagy [13]. The polyphenolic-rich pinus radiata bark extract induced caspase-independent apoptosis-like cell death in MCF-7 human breast carcinoma cells [14]. Resveratrol-rich red wine polyphenols also have been shown to cause acute lymphoblastic leukemia cells growth stagnation and promote apoptosis [15]. Blueberry fermentation products which contain enriched polyphenols inhibited proliferation, mammosphere formation, cell mobility, and cell migration potential in breast cancer cells, and reduced tumor development and metastasis in mice [16]. Trials of resveratrol have been completed on phase 1 of clinical studies in malignant hepatic tissue and confirmed its pharmacological efficacy [17]. In an intervention trial, fumonisin B1, a class 2B carcinogen, was significantly decreased in the blood and urine of people treated daily with green tea polyphenols capsules, which means the polyphenols are effective in inhibiting carcinogen-induced tumorigenic effects [18]. Collectively, polyphenols are significantly involved in cancer prevention and therapy.

Senescence is a novel anti-cancer strategy. Since it was described in 1997 that oncogenic RAS could induce senescence in rat and human primary cells with an activation of p53 or p16/retinoblastoma protein (Rb) signaling pathways, the idea of inducing premature senescence in response to abnormal mitogenic signaling to halt carcinogenesis had been proposed [19]_._ In 2005, Braig et al. confirmed senescence mediated by histone H3 lysine 9 methyltransferase SUV39H could inhibit formation of lymphoma in mice [20]. Another research demonstrated that senescence induced by activation of p53 restricted tumorigenesis caused by loss of *Pten* both in primary mouse embryonic fibroblasts and in mice. It was observed that numerous senescent cells in prostate hyperplasia/PIN were rarely in regions of carcinoma in specimens from patients with early-stage human prostate cancer [21], which further proved senescence is an initial barrier of tumorigenesis. Remarkably, cancer cells can enter a permanent cell cycle arrest if subjected to certain insults. Given that cells escape from senescence may gain the ability of unlimited proliferation and develop into cancer cells, restoring cancer cell senescence may be a positive and effective anti-cancer strategy. It is demonstrated that insulin-like growth factor binding protein 7 (IGFBP7) induced the G1/G2 cell cycle arrest and senescence to inhibit growth of triple-negative breast cancer cells both in vitro and in mice [22]. Senescence program was also restored by MiR-22, a novel senescence-associated miRNA, in cancer cells to inhibit tumor growth and metastasis in a mouse model burdened with breast carcinoma [23].

Interestingly, treatment with different concentrations of drugs on cancer cells may lead to different mechanisms of carcinoma suppression. Unlike high-dose effects of promoting apoptosis, low concentration of drugs exerts a pro-senescence effect. For instance, 10–50 mM metformin caused apoptosis confirmed by Annexin V staining and accumulation of cleaved poly ADP-ribose polymerase (PARP) protein while 1 mM metformin increased the number of senescence associated-β-galactosidase (SA-β-gal) positive cells, enhanced the protein expression level of differentiated embryo-chondrocyte expressed gene 1 (Decl), and increased the proportion of cells in G0/G1 phase in HepG2 and Bel-7402 cells. Metformin activated AMP-activated protein kinase (AMPK) at senescence-inducing concentrations, and inhibited phosphoinositide 3-kinase (PI3K)/protein kinase B (AKT)/mammalian target of rapamycin (mTOR) pathway at apoptosis inducing concentrations [24,25]. Administration of 1 μM doxorubicin induced apoptosis as demonstrated by DNA fragmentation, whereas at a lower concentration (0.1 μM) it triggered senescence in human neuroblastoma cell line SKN-SH. It is of note that activation of H2A histone family member X (H2AX), a marker for double-strand break formation, occurred mainly at apoptosis inducing drug concentrations while acetylation induction of histone H3 at the promoter of p21 only occurred at senescence-inducing concentrations [26]. Diosmin is a citrus fruit flavonoid, it mediated cytotoxic autophagy and subsequent apoptosis at high concentration (20 μM) while it induced cytostatic autophagy and subsequent senescence at low concentrations (5 and 10 μM) in breast cancer cells [27]. Nagano et al. found senescence was induced by the low doses of etoposide, whereas apoptosis was triggered at higher doses in HepG2 cells. Then they compared gene expression profiles of senescent and apoptotic cells by microarray analysis. A total of 20 genes were upregulated specifically in senescent cells, and six of them were also upregulated during replicative senescence of normal human diploid fibroblasts, suggesting that upregulation of these genes is a general phenomenon in senescence [28]. These results indicate that senescence induction requires lower drug concentrations compared to induction of apoptosis, which means that pro-senescence therapy has fewer side effects in terms of drug toxicity. As a new strategy for cancer therapy, pro-senescence has provoked considerable interest in its potential to adopt natural products that trigger cancer cell senescence for cancer treatment. In this review, we described cell senescence and focused on polyphenols and on their anti-cancer effects and molecular mechanisms for cellular senescence.

## 2. The Characteristics of Senescent Cells

Cell senescence was first described by Hayflick in 1961. Primary cells undergo three phases during the period of culture: phase I, a period of slow proliferation before the first confluent sheet conformation; phase II, during which subcultivation is required for rapid cell proliferation; phase III, cell culture terminates with the permanent loss of proliferation potential of cells (Figure 1) [29]. To explain the reasons for the state of long-term growth arrest, which is referred to as phase III, Hayflick put forward the hypothesis that the limited lifetime of diploid cell lines in vitro may be indicative of aging or senescence at the cellular level [30]. Moreover, the senescence observed by Hayflick which is associated with telomere loss or dysfunction is termed “replicative senescence”. In the absence of telomere shortening, senescence can be also induced by some stimuli, since it arises before telomere shortening. This type of senescence is termed premature (Figure 1) [31]. It was later proved that there is also cell senescence in vivo, including mouse embryo mesonephros, endolymphatic sac of the inner ear, and apical ectodermal ridge of the limbs; human maternal-fetal interface, human rheumatoid arthritis tissues, human skin, disc, and lung [32,33,34,35,36,37,38].

There are many common features in senescent cells (Figure 1), such as enlarged and irregular shaped cell body, increased activity of SA-β-gal, and cell cycle arrest. Cells in senescent states are usually accompanied by a formation of senescence-associated heterochromatin foci (SAHFs), a lack of Ki67 which is associated with cell cycle, induction of DNA damage response (DDR), and activation of p53/p21 as well as p16/Rb signaling pathway [39,40]. Apoptotic resistance is widely observed, which may be attributed to inactivation of p53 and upregulation of apoptotic B-cell lymphoma-2 (BCL-2) in senescent human fibroblasts and keratinocytes [41,42]. However, senescent endothelial cells show increased sensitivity to apoptotic with reduced BCL-2 and increased BCL-2-associated X apoptosis regulator (BAX) expression [43]. Senescent cells usually secrete a series of bioactive molecules, which is referred to as senescence-associated secretory phenotype (SASP) [44]. Secreted molecules including pro-inflammatory cytokines, such as interleukin-6 (IL-6) and IL-8, chemokines, growth factors, and matrix remodeling enzymes, can exert a series of physiological functions such as senescence promotion of adjacent cells in the tissue microenvironment [39]. In fact, not all above characteristics are commonly shared by different types of senescent cells.

## 3. Natural Polyphenols for Cancer Therapy

### 3.1. Sources of Polyphenols

Sources of polyphenols are plentiful in the natural world. Ellagic acid, gallic acid, tannic acid, and capsaicin are common phenolic acids, prevalently found in the widest variety of plant-derived foods such as teas, tropical fruits, soy, and wheat products [45]. Resveratrol (3,4’,5-trihydroxystilbene) is a representative member of the stilbene family, which has become a popular substance because of the explanation of the “French paradox”. It is abundant in our regular diets such as grape skin, peanuts, and red wine [46]. Lignans, one of the major classes of phytoestrogens, are widely distributed in *Schisandraceae* plants and other edible plants including flaxseed, sesame, cereal products, and brassica vegetables [47]. Flavonoids are secondary metabolites of plant and fungus, and high content of flavonoids may be found within common food such as fruits, vegetables, legumes, teas, dark chocolates, etc. [48].

### 3.2. Synergistic Effects of Combined Polyphenols Administration

The combination of two polyphenols has synergistic therapeutic effects (Figure 2). For example, combined administration of resveratrol and quercetin, two polyphenols, strikingly promoted the proportion of senescent cells than each single use in cultured glioma cells [49]. Polyphenols also strengthened the anti-cancer effects of traditional therapeutic regimens of chemotherapy and radiation therapy. Resveratrol enhanced senescence induced by temozolomide and further reduced temozolomide drug resistance in glioma cells [50]. Another study showed that resveratrol could re-sensitize the paclitaxel-resistant cancer cells to paclitaxel [51] and overcome gefitinib resistance of non-small cell lung cancer cells by increasing the intracellular gefitinib concentration when conjunctively used with gefitinib [52]. Resveratrol treatment also caused more DNA double-strand breaks and led to reactive oxygen species (ROS) production to reinforce ionizing radiation-induced premature senescence and reduced radioresistance in lung cancer cells [53]. The regimen of combination of doxorubicin and resveratrol enabled doxorubicin-resistant gastric cancer cells to regain doxorubicin sensitivity, reversed epithelial–mesenchymal transition and promoted cell apoptosis in vitro and achieved a tumor-inhibitory effect in nude mice bearing subcutaneous xenografts [54]. All the above cases imply the brilliant role of polyphenols in treating drug/radio-resistant stubborn cancers (Figure 2). Of note, polyphenolic compounds are part of the human diet, and recent findings suggest that the combination of curcumin and Mediterranean diet, a dietary food pattern rich in polyphenols, were effective in the management of Neurofibromatosis 1 for patients [55]. Thus, combinatorial treatment of clinical drugs or radiotherapy with resveratrol or other polyphenols can enhance their therapeutic effects and avoid their resistant effects.

### 3.3. The Molecular Mechanisms of Polyphenols for Cancer Therapy

Polyphenols participate in cancer therapy through a variety of mechanisms, including inhibition of cancer cell proliferation and migration, induction of cancer cell apoptosis, impediment of immunosuppression, etc. For instance, resveratrol inhibited the proliferation and induced apoptosis of human multiple myeloma cell lines by downregulating various proliferative and antiapoptotic gene products, including cyclin D1, cIAP-2, X-linked inhibitor of apoptosis protein (XIAP), survivin, Bcl-2, Bcl-xL, Bfl-1/A1, and TNF receptor associated factor 2 (TRAF2) [56]. A representative flavonoid quercetin is directly bound to PI3K to inhibit PI3k/AKT signaling pathway and inhibit H-RAS-induced invasion and migration in MCF10A human epithelial cells [57]. Peel polyphenolic extract and flesh polyphenolic extract from hawthorn fruit were shown to be effective for cell growth inhibition, cell-cycle arrest, and apoptosis induction via the mitochondrial pathway in MCF-7 cells [58]. Resveratrol was found to operate as a direct inhibitor of glyco-PD-L1-processing enzymes, thereby promoting the endoplasmic reticulum retention of a mannose-rich, abnormally glycosylated form of programmed death-ligand 1 (PD-L1). It was also predicted to interact with the inner surface of PD-L1 to interfere with its stability and trafficking, ultimately impeding its targeting to the cancer cell plasma membrane. When cancer cells were previously exposed to resveratrol, cytotoxic T-lymphocyte activity was notably exacerbated. These results suggest that resveratrol exerts anti-cancer property by restoring T-cell function [59].

## 4. Mechanisms of Polyphenols Targeting Senescence for Cancer Therapy and Prevention

In view of the multiple roles of senescent cells in the development of cancer and the anti-cancer properties of natural polyphenols, it is meaningful to understand the anti-cancer mechanisms of natural polyphenols targeting senescence. There are few studies focusing on lignans and phenolic acids in the literature. Thus, we herein mainly summarize the mechanisms of resveratrol and flavonoids on targeting senescence for cancer prevention and therapy (Table 1).

### 4.1. Targeting Tumor Microenvironment for Cancer Prevention

Senescent cells accumulate with age and produce large amounts of SASP factors, which may become deleterious if left unchecked. For instance, fibroblasts are the main cell types in tissue stroma and are involved in remodeling and repair processes. Once senescent, they express SASP factors, which resemble that of fibroblasts adjacent to some carcinomas, creating a tumor-favoring tissue microenvironment and promoting proliferation of melanoma cells and colorectal cancer formation [84,85]. Targeting SASP to induce an anti-tumor microenvironment may be suitable for cancer prevention. Chronic resveratrol treatment reduced secretion of SASP factors in senescent MRC5 fibroblasts, thereby mitigating intracellular signaling including phosphorylation of extracellular signal-regulated kinase (ERK) and S6 proteins in endothelial colony forming cells to reduce their angiogenesis and inhibiting epithelial-to-mesenchymal transition, proliferation, and invasiveness of melanoma cells, respectively (Figure 3) [86,87]. A plant flavone apigenin suppressed SASP secretion profile of human fibroblast cells by blunting IκBζ expression, thereby reduced the aggressive phenotype of human breast cancer cells, and oral administration of apigenin significantly inhibited SASP in the kidneys of aged rats [88,89]. In our laboratory, we demonstrated that resveratrol increased expression of silent information regulator 1 (SIRT1) to reduce SASP and inhibited age-dependent spontaneous tumorigenesis in annual fish *Nothobranchius guentheri* [90,91]. Interestingly, in a clinical trial, researchers have demonstrated that the combination of dasatinib and quercetin reduced plasma SASP factors [92]. Thus, natural polyphenols are potential in targeting tumor microenvironments to prevent cancer (Figure 3).

### 4.2. Inducing Cancer Cell Senescence for Cancer Therapy

#### 4.2.1. Oncogene-Induced Senescence (OIS)

Premature senescence induced by oncogenes is termed OIS. OIS was primitively observed in 1997 when oncogenic *RAS* was introduced into primary human diploid fibroblasts, oncogenic *RAS* can transform primary cells to a tumorigenic state in the case of tumor suppressors inactivation or the presence of cooperating oncogene. However, expression of oncogenic *RAS* resulted in a permanent G1 arrest accompanied by accumulation of p53 and p16 in primary rodent or human cells [19]. Other oncogenes or tumor suppressor genes which are able to induce senescence have been identified since then. Oncogenes such as *RAS* induced senescence are frequently found in cancer cells. Many polyphenols exert their function via OIS (Figure 3 and Figure 4). For example, suppression of the RAS/mitogen-activated protein kinase (MAPK)/ERK and PI3K/AKT signaling pathways mediated by quercetin promoted glioma cell senescence [71]. Similarly, the oncogene *AKT* was a functional target in cell cycle arrest mediated by beta-naphthoflavone, which is a synthetically derived flavonoid. The inactivation of PI3K/AKT signaling modulated by beta-naphthoflavone inhibited cyclinD1/D3 and cyclin-dependent kinase 4 (CDK4), resulted in cell cycle arrest in breast cancer cells [72]. Baicalin, a natural flavonoid glycoside, elicited human colon cancer cell senescence by activating RAS/Raf/MEK/ERK signaling via enhanced combination of hypoxia-response gene decidual protein induced by progesterone (*DEPP*) and *RAS* [73]. P53 and p16 are crucial molecules involved in senescence induction. P53 activation was required for resveratrol to upregulate C-X-C motif chemokine receptor 2 (CXCR2) to drive cancer cell senescence and protect the cells from apoptosis (Figure 2) [60]. Pterostilbene, a dimethylated analogue of resveratrol, exerted therapeutic activity on lung cancer cells by downregulating the human telomerase reverse transcriptase expression to inhibit telomerase activity, promoting DDR and S phase arrest, and subsequently inducing cell senescence in a p53-dependent manner [67,68]. In p53-null H1299 cells, p16/Rb signaling pathway exerted a positive effect in resveratrol derivative Pauciflorol B induced senescence [69]. p16/Rb pathway was also activated by DEPP in baicalin-induced senescence in colon cancer cells, and baicalin effectively inhibited tumor growth in a xenograft mouse model [73]. Not only the classical genes such as *RAS*, *AKT1*, *TP53*, and cyclin-dependent kinase inhibitor 2A (*CDKN2A*), but also deleted in liver cancer1 (*DLC1*) and human epidermal growth factor receptor 2 (*HER*2) are the main factors which are involved in OIS. *DLC1* is a potential tumor suppressor gene, previous research in our laboratory has shown resveratrol upregulated DLC1 expression to induce SASP and increase SA-β-gal positive stained breast and lung cancer cells (Figure 2) [46]. IdB 1016, which is a complex of natural flavonoid silybin with phosphatidylcholine, reduced the transcription and translation levels of proto-oncogene human epidermal growth factor receptor 2 (HER-2)/neu and induced a senescence-like phenotype both in SKBR3 cells and HER-2/neu transgenic mice. Furthermore, senescence mediated by IdB 1016 was associated with increased p53 mRNA as well as increased infiltration of neutrophils, CD4, and CD8 T-cells [74].

#### 4.2.2. Oxidative Stress-Induced Senescence

The free radical theory, also known as the oxidative stress theory, is one of the most classical mechanisms for promoting cellular senescence. ROS induced by multiple extracellular and intracellular effects is widely studied as a main cause of oxidative stress. In senescence, a variety of stressors such as endoplasmic reticulum stress and mitochondrial damage can induce the production of ROS, causing downstream DNA damage, increasing expression of molecules such as p53 and p21, and blocking cell cycle [46,75,80]. B-myb expression levels were downregulated in the aortas of aged mouse, and the inhibition of B-myb induced senescence in primary cultured human aortic endothelial cells, which is mediated by ROS production through p53/p21 signal pathway [93]. In prostate cancer cells, the loss of histone lysine methyltransferase 2D promoted senescence via DNA damage mediated by an increased intracellular ROS level [94].

ROS is also involved in the senescence process of cancer cells induced by natural polyphenols (Figure 3 and Figure 4). Low dose resveratrol markedly increased ROS production by inducing dihydronicotinamide adenine dinucleotide phosphate (NADPH) oxidase-5 and triggered senescence to inhibit the growth of non-small cell lung cancer cells [95]. Our previous results showed that resveratrol induced mitochondrial dysfunction and consequently promoted oxidative stress, then triggered senescence by upregulation of DLC1 in breast and lung cancer cells (Figure 2) [46]. Diosmin, a kind of flavonoid, increased the proportion of SA-β-gal positive cells accompanied by an elevated production of ROS, total superoxide, mitochondrial superoxide, and nitric oxide in breast cancer cell lines MCF-7 and SK-BR-3 cells [27]. Moreover, apigenin increased ROS/RNS (Reactive Nitrogen Species) generation in a dose-dependent manner, and mitochondrial superoxide was also induced to cause oxidative damage and induce senescence in human colorectal cancer cells. During oxidative stress-induced by apigenin, p16/Rb and p53-independent p21 signaling pathways were activated, cyclin E expression was downregulated by p21, while cyclin D1 was inhibited by p16 to block cell cycle and induce ultimate cell senescence [75]. Coumestrol has a structure similar to isoflavones, purified from the leaves of *Glycine max* (L.) Merrill. It inhibited the activity of protein kinase CKII to stimulate ROS production, and then activated the p53/p21 signal cascade, resulting in the induction of senescence in both breast cancer and colon cancer cells [76]. Furthermore, human neuroblastoma cell senescence could be induced by rotenone, whose structure is similar with isoflavones, and intracellular Ca^2+^ concentration was increased after treatment of rotenone, leading to the production of ROS, cell cycle arrest, and subsequent cellular senescence [77].

#### 4.2.3. DDR-Induced Senescence

A persistent DDR is usually considered as a marker of senescence, and it is regarded as the leading cause of cell senescence. DDR can be involved in the senescence process associated with telomeres. The loss of hTERT, caused shortened telomere length and DDR, and induced senescence in male germ cells [96]. Exogenous DNA-damaging agents such as ionizing can also induce senescence through a DDR pathway. For example, hSMG-1, recognized as a main member of the phosphoinositide 3-kinase-related protein kinase (PIKK) family, is involved in ionizing irradiation induced senescence. Ionizing irradiation inhibited the activity of hSMG-1 and released it from p53 pre-mRNA, boosted the binding of ribosomal protein L26 (RPL26) to p53 pre-mRNA, and recruited Serine/Arginine-rich splicing factor SRSF7, to generate alternatively spliced p53β RNA, which is a key player in DDR induced senescence [97].

Natural polyphenols are also involved in eliciting DDR-induced senescence (Figure 3 and Figure 4). 3,3’,4,4’-tetrahydroxy-trans-stilbene, a resveratrol derivative, accelerated senescence and inhibited proliferation in ovarian cancer cells, which was accompanied by the increased DNA damage and generation of ROS by reducing DNA damage repair capacity and activity of enzymatic antioxidants, respectively [70]. Senescence-like growth arrest induced by resveratrol was associated with the instability of telomeric DNA, as well as upregulation of BRCA1 in osteosarcoma cells [61]. In addition, resveratrol sequentially induced redox stress and DDR to drive senescence in p53-positive cancer cells (Figure 2) [62]. Epigallocatechin gallate, another polyphenol, is a flavonoid extracted from tea leaves. The combination of epigallocatechin gallate and sodium butyrate induced the expression of γ-H2AX, which is a marker of DDR, and caused cell cycle arrest in colon cancer cells [78].

#### 4.2.4. Endoplasmic Reticulum (ER) Stress-Induced Senescence

In addition to be the main site of calcium ion pools and lipid synthesis, the endoplasmic reticulum is also a site for the synthesis of transmembrane proteins and secreted proteins. For some stress stimuli, like mutation, oxidative stress, hypoxia, calcium imbalance, nutrient deprivation, and so on, unfolded or misfolded proteins accumulate abnormally in the endoplasmic reticulum, leading to dysfunction of the endoplasmic reticulum. These situations are called ER stress. ER stress is also caused by physiological requirements including high levels of secretory activity. The involvement of ER stress in the senescence process has been established: p38 MAPK is well known to be involved in cellular senescence, its activation triggered ER stress and resulted in senescence by upregulating unfolded protein response (UPR) sensor activating transcription factor 6 (ATF6α) in both transformed and nontransformed cells. The ER stress/ATF6α axis also mediated cellular senescence induced by RAS overexpression and UV irradiation [98].

ER stress plays an important role in polyphenol-mediated cell senescence (Figure 3 and Figure 4). An active flavonoid component oroxin A, which is obtained from the traditional Chinese herb *Oroxylum indicum* (L.) Kurz, increased ER-Tracker Red-positive cell population and upregulated ER stress-related proteins activating transcription factor 4 (ATF4) and binding immunoglobulin protein (GRP78), caused cell cycle arrest at the G2/M phase and induced senescence in human breast cancer cells while p38-specific inhibitor SB203580 blocked ER stress induced senescence [79]. Cristacarpin, a known isoflavonoid isolated from *Erythrina suberosa* stem bark, enhanced ER stress with increased expressions of GRP-94 and PERK by activation of p38, thereby generating sub-lethal ROS, elevating the expression of p21^waf1^ in a p53-independent manner, and reducing the expressions of Cdk-2 and cyclinD1, which in turn caused cellular senescence through G1 phase cell cycle arrest in pancreatic and breast cancer cells [80]. After treatment with flavokawain B, a flavonoid compound, approximately 60% of human glioblastoma multiforme cells became senescent and this result was attributed to ER stress-dependent autophagy, which was regulated by ATF-4/DNA damage inducible transcript 3 (DDIT3)/tribbles pseudokinase 3 (TRIB3)/mTOR signaling pathway [81]. Penta-1,2,3,4,6-Ogalloyl-β-d-glucose, a phenolic acid compound, triggered autophagy and activated ERN1 and EIF2AK3 arms of UPR signaling pathways to promote senescence both in cancer cells and a xenograft mouse model of human HepG2 liver cancer [82]. Cristacarpin effectively inhibited allograft mouse tumor growth by promoting ER stress-mediated ROS generation to induce premature senescence [80].

#### 4.2.5. Other Senescence Mechanisms Induced by Polyphenols

Transcription factors including forkhead box O (FOXO), MYB-related protein B (B-MYB), E2F transcription factor 1 (E2F1), DLX2, and nuclear factor kappa-light-chain-enhancer of activated B cells (NF-κB) are also involved in senescence induction. The FOXO family of transcription factors played a role in differentiation towards senescence. In primary cardiac microvascular endothelial cells, FoxO3a overexpression, significantly suppressed the senescence process by regulating the antioxidant/ROS/p27Kip1 pathway [99]. B-MYB, a member of the myeloblastosis family of transcription factors, is one of the most highly downregulated transcription factors upon senescence, and its expression could rescue RAS-induced premature senescence in mouse embryonic fibroblasts [100,101]. E2F1 belongs to a family of heterodimeric transcription factors and inhibition of E2F1 activity increased melanoma cell senescence and death for cancer therapy [102]. DLX2, a homeobox transcription factor, mediated senescence bypass phenotype in relatively old, presenescent E7 Human diploid BJ fibroblasts [103]. Polyphenols are shown to induce cancer cell senescence through regulating various transcription factors (Figure 4). As a transcription factor, NF-κB was activated with punicalagin, a polyphenol derived from pomegranate, and translocated from the cytoplasm to the nucleus, leading to subsequent senescent phenotype in papillary thyroid carcinoma BCPAP cells [104,105]. Our previous research showed that resveratrol elevated expression of SIRT1 and DLC1 to upregulate FOXO3a level, then increased NF-κB expression and promoted senescence in cancer cells [46].

In glioma cells, resveratrol induced senescence by attenuating histone H2B mono-ubiquitination at K120 [63] or decreasing the expression of POK erythroid myeloid ontogenic factor (Pokemon) via inhibiting the activity of its promoter [64], and inhibition of histone deacetylases with sodium butyrate enhanced the resveratrol-mediated pro-senescence effect [106]. Gallotannin, one of phenolic acids, activated the AMPK/SIRT1 axis to enhance SA-β-gal activity in hepatocellular carcinoma cells and inhibited the growth of HepG2 cells implanted in nude mice [83]. Another in vivo experiment has confirmed that resveratrol induced senescence in gastric cancer cell xenografts from nude mice and inhibited growth of tumor in a SIRT1-dependent manner [65]. Downregulation of rapamycin-insensitive companion of mammalian target of rapamycin (Rictor), a subunit of mTORC2, may be associated with resveratrol-mediated senescence induction in vitro as well as suppression of UV-induced skin carcinogenesis in vivo [66].

## 5. Conclusions

Polyphenols are ubiquitous in our diet and are an important source of anti-cancer compounds. Targeting senescence is an emerging anti-cancer strategy and polyphenols have a role in senescence regulation. This review is valuable in terms of clarifying mechanisms of polyphenols targeting senescence for cancer prevention and therapy. We concluded that polyphenols inhibit SASP factors secretion from senescent cells to form anti-tumor microenvironment to prevent cancer. Once tumorgenesis occurs, polyphenols are able to inhibit cancer by the mechanisms of oncogene-, oxidative stress-, DDR-, and ER stress-induced cancer cell senescence (Figure 3 and Figure 4). Concomitant usage of two polyphenols or a mixture of polyphenols and other anti-cancer drugs enhances therapeutic efficiency. Combined use of polyphenols or taking polyphenols as auxiliary compounds clinically will be a key consideration in the future.

## Figures and Tables

**Figure 1 ijms-21-00684-f001:**
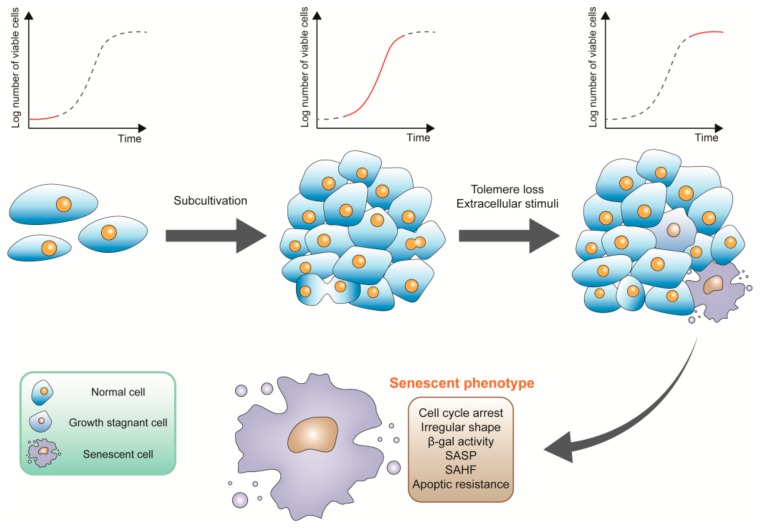
Process and phenotype of senescence. Cell growth curve, senescence process, and characteristics of senescent cells are displayed. SASP; senescence-associated secretory phenotype SAHF, senescence-associated heterochromatin foci.

**Figure 2 ijms-21-00684-f002:**
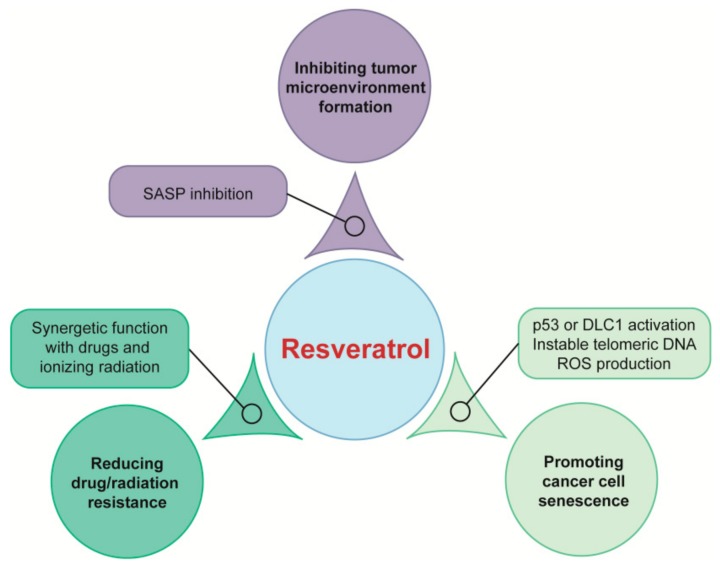
Potential functions of resveratrol in anti-tumor therapy. Resveratrol inhibit tumor microenvironment for cancer prevention, reduce drug/radiation resistance and induce cancer cell senescence for cancer therapy. SASP, senescence-associated secretory phenotype; DLC1, deleted in liver cancer1; ROS, reactive oxygen species.

**Figure 3 ijms-21-00684-f003:**
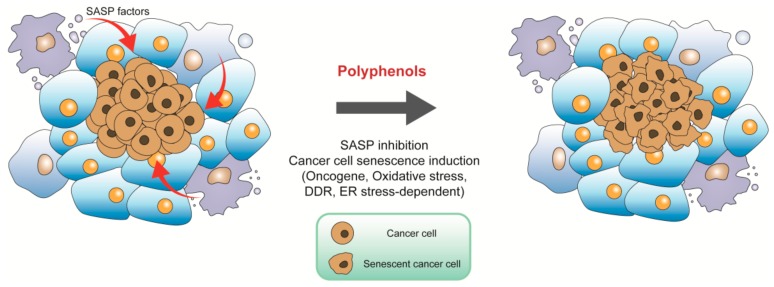
Roles of polyphenols in tumor suppression. Polyphenols inhibit secretion factors from senescent cells to impair tumor microenvironment and promote oncogene-, oxidative stress-, DNA damage response (DDR)-, and endoplasmic reticulum (ER) stress-induced cancer cell senescence.

**Figure 4 ijms-21-00684-f004:**
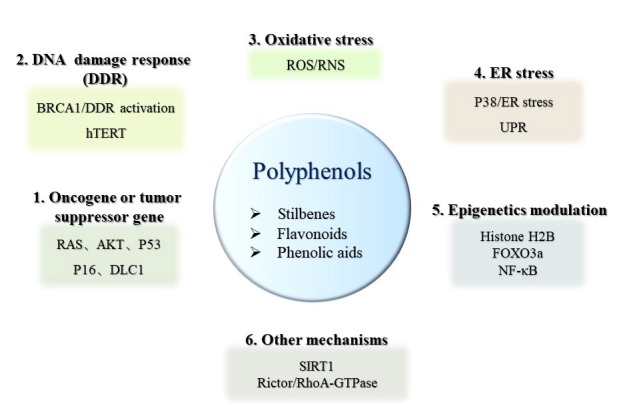
Molecular targets of polyphenols in senescence induction. Polyphenols can induce cancer cell senescence via various pathways, including (**1**) regulation of oncogene or tumor suppressor gene expression, (**2**) activation of DDR, (**3**) promotion of oxidative stress, (**4**) induction of ER stress, (**5**) modulation of epigenetics, and (**6**) other mechanisms. AKT, protein kinase B; DLC1, deleted in liver cancer1; hTERT, human telomerase reverse transcriptase; ROS, reactive oxygen species; RNS, reactive nitrogen species; ER stress, endoplasmic reticulum stress; UPR, unfolded protein response; FOXO3a, forkhead box O 3a; NF-κB, nuclear factor kappa-light-chain-enhancer of activated B cells.

**Table 1 ijms-21-00684-t001:** Polyphenols and polyphenol derivatives as cancer cell senescence inducers and their effect on signal transduction pathways.

Classification	Compounds	Concentration	Pathways	Cancer Type	Reference
Resveratrol and its derivatives	Resveratrol	25/50 (μM)	p53/CXCR2	Osteosarcoma Fibrosarcoma Lung cancer	[60]
		50 (μM)	BRCA1/DDR	Osteosarcoma	[61]
		30 (μM)	ROS/DDR	Colon cancer	[62]
		100 (μM)	ROS/DLC1/SASP	Breast cancer Lung cancer	[46]
		6/20 (μM) 100 (μM)	Histone H2B Pokemon	Glioma	[63,64]
		25/50 (μM)	SIRT1	Gastric cancer	[65]
		50 (μM)	Rictor/RhoA-GTPase	Nonmelanoma skin cancer	[66]
	Pterostilbene	2.5/5/50 (μM)	hTERT/DDR	Lung cancer	[67,68]
	Pauciflorol B	10 (μM)	p16/Rb	Lung cancer	[69]
	3,3’,4,4’-tetrahydroxy-trans-stilbene	10/50/100 (μM)	ROS DDR	Ovarian cancer	[70]
Flavonoids	Quercetin	50/100/200 (μM)	RAS/MAPK/ERK PI3K/AKT	Glioma	[71]
	Beta-naphthoflavone	10 (μM)	PI3K/AKT/cyclinD1/D3 MAPK/ERK	Breast cancer	[72]
	Baicalin	10/20/40 (μM)	DEPP/RAS/Raf/MEK/ERK DEPP/p16/Rb	Colon cancer	[73]
	IdB 1016	63.2/126.5 (μg/mL)	HER-2/neu p53	Breast cancer	[74]
	Diosmin	5/10 (μM)	ROS DDR	Breast cancer	[27]
	Apigenin	Above 25 (μM)	ROS/RNS p16/cyclin D1/p-Rb p21/cyclin E/p-Rb	Colorectal cancer	[75]
	Coumestrol	50 (μM)	CKII/ROS/p53/p21	Breast cancer Colon cancer	[76]
	Rotenone	0.4 (μM)	Ca^2+^/ROS	Neuroblastoma	[77]
	Epigallocatechin gallate	10 (μM)	DDR	Colon cancer	[78]
	Oroxin A	5/10/15/20 (μM)	p38/ER stress	Breast cancer	[79]
	Cristacarpin	1 (μM)	p38/ER stress/ROS/p21	Pancreatic Breast cancer	[80]
	Flavokawain B	3 (μg/mL)	ATF4/DDIT3/TRIB3/AkT/mTOR	Glioblastoma multiforme	[81]
Phenolic acids	Penta-1,2,3,4,6-Ogalloyl-β-d-glucose	25 (μM)	UPR	Liver cancer Breast cancer Lung cancer	[82]
	Gallotannin	40 (μM)	AMPK/SIRT1	Hepatocellular carcinoma	[83]

CXCR2: C-X-C motif chemokine receptor 2; DDR: DNA damage response; ROS: reactive oxygen species; DLC1: deleted in liver cancer1; SASP: senescence-associated secretory phenotype; Pokemon: POK erythroid myeloid ontogenic factor; SIRT1: silent information regulator 1; Rictor: rapamycin-insensitive companion of mammalian target of rapamycin; hTERT: human telomerase reverse transcriptase; Rb: retinoblastoma protein; MAPK: mitogen-activated protein kinase; ERK: extracellular signal-regulated kinase; PI3K: phosphoinositide 3-kinase; AKT: protein kinase B; DEPP: decidual protein induced by progesterone; HER-2/neu: human epidermal growth factor receptor 2; RNS: reactive nitrogen species; ER stress: endoplasmic reticulum stress; ATF4: activating transcription factor 4; DDIT3: DNA damage inducible transcript 3; TRIB3: tribbles pseudokinase 3; mTOR: mammalian target of rapamycin; UPR: unfolded protein response; AMPK: AMP-activated protein kinase.

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
