# Peer review of "Natural Polyphenols Targeting Senescence: A Novel Prevention and Therapy Strategy for Cancer"

_ijms, 2020, doi:10.3390/ijms21020684_

Reviewer 1 Report

The review entitled ''Natural Polyphenols Targeting Senescence: A Novel Prevention and Therapy Strategy for Cancer'' describes the effects of Polyphenols on senescence for targeting cancer. I found that it is a good work and I  suggest some comments to improve this work. The authors have shown general effects of polyphenols on senescence without
treating the signaling pathways leading to this phenomenon. I invite others to add data concerning the specific targeting of the transcription factors involved in differentiation towards senescence.

Author Response

Response to Reviewer 1 Comments

Question:

The review entitled ''Natural Polyphenols Targeting Senescence: A Novel Prevention and Therapy Strategy for Cancer'' describes the effects of Polyphenols on senescence for targeting cancer. I found that it is a good work and I suggest some comments to improve this work. The authors have shown general effects of polyphenols on senescence without treating the signaling pathways leading to this phenomenon. I invite others to add data concerning the specific targeting of the transcription factors involved in differentiation towards senescence.

Response: Thank you for your encouragement and suggestion. We have added data concerning the transcription factors involved in differentiation towards senescence in line 309 to 325. The transcription factors FOXO, B-MYB, E2F1, DLX2 and NF-κB involved in senescence differentiation. In primary cardiac microvascular endothelial cells, FoxO3a overexpression significantly suppressed the senescence process by regulating the antioxidant/ROS/p27Kip1 pathway (Qi, X. F et al., 2015). Expression of B-MYB could rescue RAS-induced premature senescence in mouse embryonic fibroblasts (Mowla, S.N et al., 2014; Masselink,H et al. 2000). Inhibition of E2F1 activity increased melanoma cell senescence and death for cancer therapy (Rouaud, F et al., 2018). DLX2 mediated senescence bypass phenotype in relatively old, presenescent E7 Human diploid BJ fibroblasts (Wang, Y et al., 2016;). For polyphenols, punicalagin activated transcription factors NF-κB and promoted its translocation from the cytoplasm to the nucleus, leading to papillary thyroid carcinoma BCPAP cell senescence (Cheng, X et al., 2018). Resveratrol elevated expression of SIRT1 and DLC1 to upregulate FOXO3a level, then increased NF-κB expression, and promoted senescence in cancer cells (Ji, S et al., 2018). These modifications are marked with purple font.

Reviewer 2 Report

The authors discuss the role of senecscence in cancer and the possibility to target senescence for therapeutic purposes by using polyphenols.

To provide a comprehensive overview of polyphenols action, I recommend the authors to mention, in the Introduction session, additional mechanisms of polyphenols which are unrelated to senescence, like miRNAs modulation, epigenetics modulation both in solid and hematological malignancies.

The following literature is recommended:

- Raimondi L et al., Curr Med Chem. 2018 Jun 29. doi: 10.2174/0929867325666180629153141

-Arora I, Sharma M, Tollefsbol TO, Int J Mol Si 2019

-Shukla S, Penta D, Mondal P, Meeran SM, Adv Exp Med Biol. 2019

-Raimondi L et al., Cancers (Basel). 2019 Jul 16;11(7). pii: E990. doi: 10.3390/cancers11070990.

Author Response

Response to Reviewer 2 Comments

Question: The authors discuss the role of senescence in cancer and the possibility to target senescence for therapeutic purposes by using polyphenols.

To provide a comprehensive overview of polyphenols action, I recommend the authors to mention, in the Introduction session, additional mechanisms of polyphenols which are unrelated to senescence, like miRNAs modulation, epigenetics modulation both in solid and hematological malignancies.

The following literature is recommended:

- Raimondi L et al., Curr Med Chem. 2018 Jun 29. doi: 10.2174/0929867325666180629153141

-Arora I, Sharma M, Tollefsbol TO, Int J Mol Si 2019

-Shukla S, Penta D, Mondal P, Meeran SM, Adv Exp Med Biol. 2019

-Raimondi L et al., Cancers (Basel). 2019 Jul 16;11(7). pii: E990. doi: 10.3390/cancers11070990.

Response: Thank you for your valuable suggestions and the references you provided. We did mention some additional anti-cancer mechanisms of polyphenols on apoptosis, autophagy, proliferation and migration in “3.1. The anticancer properties of polyphenols”, which are unrelated to cell senescence, but they are not comprehensive. Based on your suggestions and the references, we have added miRNAs modulation and epigenetics modulation effects of polyphenols in solid and hematological malignancies in line 40 to 48, together with the mechanisms of polyphenols on apoptosis, autophagy, proliferation and migration (moved from line115 to 130 to line 48 to 62) in the Introduction session. Due to the shift of polyphenol mechanisms on apoptosis, autophagy, proliferation and migration in 3.1, the sentence “Polyphenols have a variety of beneficial effects on the prevention of chronic diseases because of their anti-oxidation and anti-inflammatory ability” has been deleted to make the context fluent, and “3.1. The anticancer properties of polyphenols” has been modified to “3.1 Sources of polyphenols" in line 125. We have also modified “The tumor suppressor effects of polyphenols are ascribed to multiple pathways” with “The mechanisms include epigenetic modulation as well” in line 41. These modifications are marked with green font.

Reviewer 3 Report

I withdraw from reviewing this

For me as clinician this manuscript is impossible to read and to understand. It has first to be reviewed by a native speaker as there are numerous grammatical errors with sentences lacking verbs and missing other terms.

in every paragraph, data on cell lines, animal experiments and observations from clinical studies are mixed together - and this is extremely misleading and exhausting

The authors should also be careful with their statements that observations in cell cultures have an impact on cancer prevention ( in cell culture? in animals or in humans???) or even in cancer therapy

Author Response

Response to Reviewer 3 Comments

Question: I withdraw from reviewing this.

For me as clinician this manuscript is impossible to read and to understand. It has first to be reviewed by a native speaker as there are numerous grammatical errors with sentences lacking verbs and missing other terms.

in every paragraph, data on cell lines, animal experiments and observations from clinical studies are mixed together - and this is extremely misleading and exhausting.

The authors should also be careful with their statements that observations in cell cultures have an impact on cancer prevention (in cell culture? in animals or in humans???) or even in cancer therapy.

Response: Thank you for your comments. In this manuscript, we reviewed natural polyphenols targeting senescence as a novel prevention and therapy strategy for cancer, and summarized the related molecular mechanisms in each section on the types of polyphenols. According to your suggestions, we have made the following modifications in blue font.

(1)        In line 53 to 57, we have changed sentence order, which acute lymphoblastic leukaemia cells were described and then tumor development and metastasis in mice.

(2)        In line 86, we have changed “1 μM” with “Administration of 1 μM”.

(3)        In line 140, we have changed “enhance” with “strengthened”.

(4)        In line 152, we have added “and”.

(5)        In line 154, we have added “for patients”.

(6)       In line 179, we have changed “by blunting IκBζ expression to reduce” with “by blunting IκBζ expression, thereby reduced”.

(7)        In line 197, we have changed “human or rodent” with “rodent or human”.

(8)        In line 244, we have added “and”, and have moved “in human colorectal cancer cells” to the end of the sentence.

(9)        In line 252 and 253, we have changed “human neuroblastoma senescence” with “human neuroblastoma cell senescence” and have added “and” to link the two sentences.

(10)      In line 298, We have moved “After treatment with flavokawain B” to the beginning of the sentence and added “which” to link the two sentences.

(11)      We have moved the sentence “Penta-1,2,3,4,6-Ogalloyl-β-D-glucose, a phenolic acid compound, triggered autophagy and activated ERN1 and EIF2AK3 arms of unfolded protein response (UPR) signaling pathways to promote senescence both in cancer cells and a xenograft mouse model of human HepG2 liver cancer” from line 284 to 287 to line 302 to 305, to keep the order from cell line to animal level.

(12)      In line 305, we have deleted “In vivo experiment showed that” and changed “cristacarpin effectively inhibited allograft mouse tumor growth by inducing senescence in tumor tissue” with “Cristacarpin effectively inhibited allograft mouse tumor growth by promoting ER stress-mediated ROS generation to induce premature senescence” to highlight the mechanism mediated via ER stress.

(13)      In line 332, we have changed “have” with “has”.